# FONDUE: AN ALGORITHM TO AUTOMATICALLY FIND THE DIMENSIONALITY OF THE LATENT REPRESENTATIONS OF VARIATIONAL AUTOENCODERS

## ABSTRACT

When training a variational autoencoder (VAE) on a given dataset, determining the number of latent variables is mostly done by grid search — a costly process in terms of computational time and carbon footprint. In this paper, we explore the intrinsic dimension estimates (IDEs) of the data and latent representations learned by VAEs. We show that the discrepancies between the IDE of the mean and sampled representations of a VAE after only a few steps of training reveal the presence of passive variables in the latent space, which, in well-behaved VAEs, indicates a superfluous number of dimensions. Using this property, we propose FONDUE: an algorithm which quickly finds the number of latent dimensions after which the mean and sampled representations start to diverge (i.e., when passive variables are introduced), providing a principled method for selecting the number of latent dimensions for VAEs and autoencoders.

## 1 INTRODUCTION

"How many latent variables should I use for this model?" is a question that many practitioners using variational autoencoders (VAEs) or autoencoders (AEs) have to deal with. When the task has been studied before, this information is available in the literature for the specific architecture and dataset used. However, when it has not, answering this question becomes more complicated. Indeed, the dimensionality of the latent representation is currently determined empirically by increasing the number of latent dimensions until the reconstruction loss or accuracy on a downstream task does not improve anymore (Doersch, 2016; Mai Ngoc & Hwang, 2020). This is a costly process requiring to fully train multiple models, and increasing the carbon footprint and time needed for an experiment.

One could wonder if it would be sufficient to use a very large number of latent dimensions in all cases. However, beside defeating the purpose of learning compressed representations, this may lead to a range of issues. For example, one would obtain lower accuracy on downstream tasks (Mai Ngoc & Hwang, 2020) and — if the number of dimensions is sufficiently large — very high reconstruction loss (Doersch, 2016). This would also hinder the interpretability of downstream task models such as linear regression, prevent investigating the learned representation with latent traversal, and increase the correlation of the latent representations (Bonheme & Grzes, 2021).

Intrinsic dimension (ID) estimation — the estimation of the minimum number of variables needed to describe the data — is an active area of research in topology, and various estimation methods have been proposed (Facco et al., 2017; Levina & Bickel, 2004). In recent years, these techniques have successfully been applied to deep learning to empirically show that the intrinsic dimension of images was much lower than their extrinsic dimension (i.e., the number of pixels) (Gong et al., 2019; Ansuini et al., 2019; Pope et al., 2021), and that the ID estimates (IDEs) of neural network classifiers with good generalisation tended to first increase, then decrease until reaching a very low IDE in the last layer (Ansuini et al., 2019). However, to the best of our knowledge, ID estimation techniques have never been applied to VAEs.

After exploring the IDEs of the representations learned by VAEs at different layers, we will show that by combining this technique with knowledge of the properties of VAEs, we can design a simple yet efficient algorithm which fulfills the criteria of the current methods (i.e., low reconstruction loss, high accuracy on downstream tasks), without requiring to fully train multiple models.

Our contributions are as follows:

(i) We provide an experimental study of the IDEs of VAEs, and found that (1) the layers of VAEs reach stable IDEs very early in the training; and (2) the IDE of the mean and sampled representations is different when some latent variables collapse.

(ii) Based on these observation we propose FONDUE: an algorithm which automatically finds the number of latent dimensions that leads to a low reconstruction loss and good accuracy. In opposition to current methods (Doersch, 2016; Mai Ngoc & Hwang, 2020), it does not require human supervision or multiple model training.

(iii) To foster reproducibility, we have released more than 35,000 ID estimates (`https://t.ly/8r3N`[1]). The library created for this experiment (`t.ly/Oh7s`) can be reused with other ID estimation techniques or models for further research in the domain.

## 2 BACKGROUND

### 2.1 VARIATIONAL AUTOENCODERS

VAEs (Kingma & Welling, 2014; Rezende & Mohamed, 2015) are deep probabilistic generative models based on variational inference. The encoder maps an input $\mathbf{x}$ to a latent representation $\mathbf{z}$, and the decoder attempts to reconstruct $\mathbf{x}$ using $\mathbf{z}$. This can be optimised by maximising $\mathcal{L}$, the evidence lower bound (ELBO)

$$\mathcal{L}(\boldsymbol{\theta}, \boldsymbol{\phi}; \mathbf{x}) = \underbrace{\mathbb{E}_{q_{\boldsymbol{\phi}}(\mathbf{z}|\mathbf{x})}[\log p_{\boldsymbol{\theta}}(\mathbf{x}|\mathbf{z})]}_{\text{reconstruction term}} - \underbrace{D_{\mathrm{KL}}\left(q_{\boldsymbol{\phi}}(\mathbf{z}|\mathbf{x})||p(\mathbf{z})\right)}_{\text{regularisation term}}, \tag{1}$$

where $p(\mathbf{z})$ is generally modelled as a standard multivariate Gaussian distribution $\mathcal{N}(0, \boldsymbol{I})$ to permit a closed form computation of the regularisation term (Doersch, 2016). The regularisation term can be further penalised by a weight $\beta$ (Higgins et al., 2017) such that

$$\mathcal{L}(\boldsymbol{\theta}, \boldsymbol{\phi}; \mathbf{x}) = \underbrace{\mathbb{E}_{q_{\boldsymbol{\phi}}(\mathbf{z}|\mathbf{x})}[\log p_{\boldsymbol{\theta}}(\mathbf{x}|\mathbf{z})]}_{\text{reconstruction term}} - \underbrace{\beta D_{\mathrm{KL}}\left(q_{\boldsymbol{\phi}}(\mathbf{z}|\mathbf{x})||p(\mathbf{z})\right)}_{\text{regularisation term}}, \tag{2}$$

reducing to equation 1 when $\beta = 1$ and to a deterministic autoencoder (AE) when $\beta = 0$. Note that we mention AEs here as a way of explaining $\beta$, and refer the reader to Appendix J for an overview of AEs.

**Polarised regime** When $\beta \geqslant 1$, VAEs are encouraged to have a high precision (i.e., low variance) on the latent variables that they use (the active variables) while maintaining the remaining — passive — variables close to $\mathcal{N}(0, \boldsymbol{I})$ (Rolinek et al., 2019). This behaviour typical for VAEs is known as polarised regime or posterior collapse and is necessary for VAEs to provide good reconstruction (Dai & Wipf, 2018; Dai et al., 2020). Because they are close to a standard Gaussian distribution, the passive variables can be used by the model to lower the KL divergence and compensate the increased divergence generated by the active variables. Moreover, their mean representation will generally be close to zero regardless of the input while their sampled representation will have a variance close to 1 (Bonheme & Grzes, 2021) (see also Appendices E and F).

### 2.2 INTRINSIC DIMENSION ESTIMATION

It is generally assumed that a dataset $\boldsymbol{X}$ of $m$ i.i.d. data examples $\boldsymbol{X}_i \in \mathbb{R}^n$ is a locally smooth non-linear transformation $g$ of a lower-dimensional dataset $\boldsymbol{Y}$ of $m$ i.i.d. samples $\boldsymbol{Y}_i \in \mathbb{R}^d$, where $d \leqslant n$ (Campadelli et al., 2015; Chollet, 2021). The goal of ID estimation is to recover $d$ given $\boldsymbol{X}$. In this section, we will detail two ID estimation techniques which use the statistical properties of the neighbourhood of each data point to estimate $d$, and provide good results for approximating the ID of deep neural network representations and deep learning datasets (Ansuini et al., 2019; Gong et al., 2019; Pope et al., 2021). See Appendix H and Campadelli et al. (2015) for more details on ID estimation techniques.

---

[1]Due to their size and to preserve anonymity, the 300 models trained for this experiment will be released after the review.

**Maximum Likelihood Estimation** Levina & Bickel (2004) modelled the neighbourhood of a given point $\boldsymbol{X}_i$ as a Poisson process in a d-dimensional sphere $S_{\boldsymbol{X}_i}(R)$ of radius $R$ around $\boldsymbol{X}_i$. This Poisson process is denoted $\{N(t, \boldsymbol{X}_i), 0 \leqslant t \leqslant R\}$, where $N(t, \boldsymbol{X}_i)$ is a random variable representing the number of neighbours of $\boldsymbol{X}_i$ within a radius $t$, and is distributed according to a Poisson distribution[2]. Each point $\boldsymbol{X}_j \in S_{\boldsymbol{X}_i}(R)$ is thus considered as an event, its arrival time $t = T(\boldsymbol{X}_i, \boldsymbol{X}_j)$ being the Euclidean distance from $\boldsymbol{X}_i$ to its $j^{th}$ neighbour $\boldsymbol{X}_j$. By expressing the rate $\lambda(t, \boldsymbol{X}_i)$ of the process $N(t, \boldsymbol{X}_i)$ as a function of the surface area of the sphere — and thus relating $\lambda(t, \boldsymbol{X}_i)$ to $d$ — they obtain a maximum likelihood estimation (MLE) of the ID $d$:

$$\bar{d}_R(\boldsymbol{X}_i) = \left[ \frac{1}{N(R, \boldsymbol{X}_i)} \sum_{j=1}^{N(R, \boldsymbol{X}_i)} \log \frac{R}{T(\boldsymbol{X}_i, \boldsymbol{X}_j)} \right]^{-1}. \tag{3}$$

Equation 3 is then simplified by fixing the number of neighbours, $k$, instead of the radius $R$ of the sphere, such that

$$\bar{d}_k(\boldsymbol{X}_i) = \left[ \frac{1}{k-1} \sum_{j=1}^{k-1} \log \frac{T(\boldsymbol{X}_i, \boldsymbol{X}_k)}{T(\boldsymbol{X}_i, \boldsymbol{X}_j)} \right]^{-1}, \tag{4}$$

where the last summand is omitted, as it is zero for $j = k$. The final IDE $\bar{d}_k$ is the averaged score over $n$ data examples (Levina & Bickel, 2004)

$$\bar{d}_k = \frac{1}{n} \sum_{i=1}^{n} \bar{d}_k(\boldsymbol{X}_i). \tag{5}$$

To obtain an accurate estimation of the ID with MLE, it is very important to choose a sufficient number of neighbours $k$ to form a dense small sphere (Levina & Bickel, 2004). On one hand, if $k$ is too small, MLE will generally underestimate the ID, and suffer from high variance (Levina & Bickel, 2004; Campadelli et al., 2015; Pope et al., 2021). On the other hand, if $k$ is too large, the ID will be overestimated (Levina & Bickel, 2004; Pope et al., 2021).

**TwoNN** Facco et al. (2017) proposed an estimation of the ID based on the ratio of the two nearest neighbours of $\boldsymbol{X}_i$, $r_{\boldsymbol{X}_i} = \frac{T(\boldsymbol{X}_i, \boldsymbol{X}_l)}{T(\boldsymbol{X}_i, \boldsymbol{X}_j)}$, where $\boldsymbol{X}_j$ and $\boldsymbol{X}_l$ are the first and second closest neighbours of $\boldsymbol{X}_i$, respectively. $r$ follows a Pareto distribution with scale $s = 1$ and shape $d$, and its density function $f(r)$ is

$$f(r) = \frac{ds^d}{r^{d+1}} = dr^{-(d+1)}. \tag{6}$$

Its cumulative distribution function is thus

$$F(r) = 1 - \frac{s^d}{r^d} = 1 - r^{-d}, \tag{7}$$

and, using Equation 7, one can readily obtain $d = \frac{-\log(1 - F(r))}{\log r}$. From this, we can see that $d$ is the slope of the straight line passing through the origin, which is given by the set of coordinates $\mathbb{S} = \{(\log r_{\boldsymbol{X}_i}, -\log(1 - F(r_{\boldsymbol{X}_i}))) \mid i = 1, \cdots, m\}$, and can be recovered by linear regression.

As TwoNN uses only two neighbours, it can be sensitive to outliers (Facco et al., 2017) and do not perform well on high ID (Pope et al., 2021), overestimating the ID in both cases.

**Ensuring an accurate analysis** Given the limitations previously mentioned, we take two remedial actions to guarantee that our analysis is as accurate as possible. To provide an IDE which is as accurate as possible with MLE, we will measure the MLE with an increasing number of neighbours and, similar to Karbauskaitė et al. (2011), retain the IDE which is stable for the largest number of $k$ values. TwoNN will be used as a complementary metric to validate our choice of $k$ for MLE. In case of significant discrepancies with a significantly higher TwoNN IDE, we will rely on the results provided by MLE.

---

[2]Note that this does not imply any distributional assumption about the dataset

### 2.3 RELATED WORK

To the best of our knowledge, the literature on finding an appropriate number of latent dimensions for VAEs is limited, and existing techniques always rely on the elbow method (i.e., visually finding where a curve "bends") (James et al., 2013).

**Comparing reconstruction error**   Doersch (2016) trained multiple models with different numbers of latent dimensions and selected the ones with the lowest reconstruction error. They noted that models performance were noticeably worse when using extreme numbers of latent dimensions. In their experiment, this happened for $|z| < 4$ and $|z| > 10,000$ for MNIST.

**Comparing accuracy on downstream tasks**   Mai Ngoc & Hwang (2020) suggested to train multiple models with different number of latent dimensions, then compare the accuracy of the latent representations on a downstream task. They observed that while a higher number of latent dimensions could lead to a lower reconstruction error, it generally caused instability on downstream tasks. They thus concluded that the best number of latent dimensions for VAEs should be the one with the least classification variance and highest accuracy, and obtained similar results for AEs.

## 3   EXPERIMENTAL SETUP

We will first investigate the IDEs of the representations learned by VAEs to show that they can be used to determine the number of latent dimensions of VAEs. We will then analyse these results in Section 4.2 and use them to design an algorithm to Facilely Obtain the Number of latent Dimensions by Unsupervised Estimation (FONDUE) in Section 4.3.

We will then assess the performance of FONDUE and compare it with the existing techniques of dimension selection discussed in Section 2.3. This will be done by ensuring that the number of dimensions selected by the elbow method for reconstruction and downstream tasks is consistent with the value proposed by FONDUE.

**Datasets**   We use three datasets with an increasing number of intrinsic dimensions: Symmetric solids (Murphy et al., 2021), dSprites (Higgins et al., 2017), and Celeba (Liu et al., 2015). The numbers of generative factors of the first two datasets are 2 and 5, respectively, and the IDE of these two datasets should be close to these values. While we do not know the generative factors of Celeba, Pope et al. (2021) reported an IDE greater than 20, which is high enough for our experiment.

**Data preprocessing**   Each image is resized to $64 \times 64 \times c$, where $c = 1$ for Symmetric solids and dSprites, and $c = 3$ for Celeba. We also removed duplicate images (i.e., cases where different rotations resulted in the same image) and labels from Symmetric solids and created a reduced version: `symsol_reduced` which is available at `https://t.ly/_CdH`.

**VAE training**   We use the $\beta$-VAE architecture detailed in Higgins et al. (2017) for all the datasets, together with the standard learning objective of VAEs, as presented in Equation 1. Each VAE is trained 5 times with a number of latent dimensions $n = 3, 6, 8, 10, 12, 18, 24, 32$ on every dataset. For Celeba, which has the highest IDE, we additionally train VAEs with latent dimensions $n = 42, 52, 62, 100, 150, 200$.

**Estimations of the ID**   For all the models, we estimate the ID of the layers' activations using 10,000 data examples each. As in Pope et al. (2021), the MLE scores are computed with $k = 3, 5, 10, 20$. Moreover, we repeat the MLE computations 3 times with different seeds to detect any variance in estimates.

**Downstream tasks**   To monitor how good the learned latent representations are on downstream tasks, we use the shape attribute of Symsol and dSprites for multi-class classification, and the 40 binary attributes of Celeba for multi-label classification. We train a logistic regression model for each task and evaluate the results with macro-F1 scores.

Additional details on our implementation can be found in Appendix C and our code is available at `t.ly/Oh7s`.

## 4 RESULTS

In this section, we will analyse the results of the experiment detailed in Section 3. First, we will review the IDE of the different datasets in Section 4.1. Then, in Section 4.2, we will discuss the variation of IDEs between different layers of VAEs when we change the number of latent dimensions and how it evolves during training. Finally, based on the findings of these sections, we propose an algorithm to Facilely Obtain the Number of latent Dimensions by Unsupervised Estimation (FONDUE) in Section 4.3.

### 4.1 ESTIMATING THE INTRINSIC DIMENSIONS OF THE DATASETS

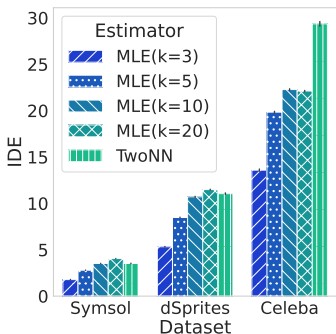

Figure 1: IDEs of dSprites, Celeba, and Symsol using different ID estimation methods.

Following Karbauskaitė et al. (2011), we will retain for our analysis the MLE estimates which are stable for the largest number of $k$ values, as detailed in Section 2.2. We can see in Figure 1 that the MLE estimations become stable when $k$ is between 10 and 20, similar to what was reported by Levina & Bickel (2004). These IDEs are also generally close to TwoNN estimations, except for Celeba, where TwoNN seems to overestimate the ID, as previously reported by Pope et al. (2021). In the rest of this paper, we will thus consider the IDEs obtained from MLE with $k = 20$ as our most likely IDEs.

As mentioned in Section 3, we have selected 3 datasets of increasing intrinsic dimensionality: Symsol (Murphy et al., 2021), dSprites (Higgins et al., 2017), and Celeba (Liu et al., 2015). Celeba's IDE was previously estimated to be 26 for MLE with $k = 20$ (Pope et al., 2021), and we know that Symsol and dSprites have 2 and 5 generative factors, respectively. We thus expect their IDEs to be close to these values. We can see in Figure 1 that MLE and TwoNN overestimate the IDs of Symsol and dSprites, with IDEs of 4 and 11 instead of the expected 2 and 5. Our result for Celeba is close to Pope et al. (2021) with an estimate of 22; the slight difference may be attributed to the difference in our averaging process (Pope et al. (2021) used the averaging described by MacKay & Ghahramani (2005) instead of the original averaging of Levina & Bickel (2004) given in Equation 5).

Overall, we can see that we get an upper bound on the true ID of the data for the datasets whose ID are known. However, we show experimentally in Appendix I that the overestimation of the data ID does not have any negative impact on our results.

### 4.2 ANALYSING THE IDEs OF THE DIFFERENT LAYERS OF VAEs

Now that we have an IDE for each of the datasets, we are interested in observing how the IDEs of VAEs' representations vary between layers and epochs, when they are trained with different numbers of latent variables (see Appendix H for additional observations).

**Mean and sampled representations have different IDEs** Looking into the IDEs of mean and sampled representations in Figure 2, we see a clear pattern emerge: when increasing the number of latent variables the IDEs remain similar up to a point, then abruptly diverge. As discussed in Section 2.1, once a VAE has enough latent variables to encode the information needed by the decoder,

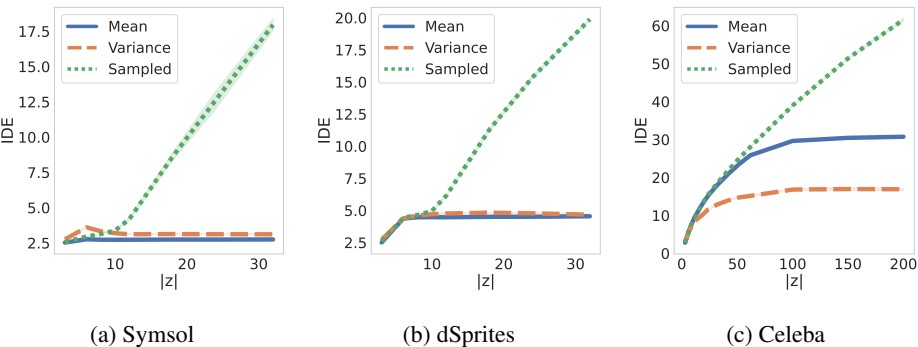

(a) Symsol        (b) dSprites        (c) Celeba

Figure 2: IDE of the mean, variance, and sampled representations of VAEs trained with an increasing number of latent dimensions $|\mathbf{z}|$. (a), (b), and (c) shows the results on Symsol, dSprites, and Celeba, respectively.

the remaining variables will become passive to minimise the KL divergence in Equation 2. This phenomenon will naturally occur when we increase the number of latent variables. Bonheme & Grzes (2021) observed that, in the context of the polarised regime, passive variables were very different in mean and sampled representations. Indeed, for sampled representations, the set of passive variables will be sampled from $\mathcal{N}(0, I)$ where they will stay close to 0 with very low variance in mean representations. They also introduced the concept of mixed variables — variables that are passive only for some data examples — and shown that they were also leading to different mean and sampled representations, albeit to a minor extent. We can thus hypothesise that the difference between the mean and sampled IDEs grows with the number of mixed and passive variables. This is verified by computing the number of active, mixed, and passive variables using the method of Bonheme & Grzes (2021), as shown in Figure 3.

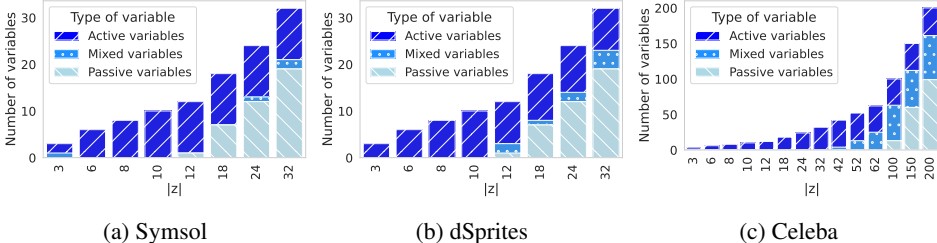

(a) Symsol        (b) dSprites        (c) Celeba

Figure 3: Quantity of active, mixed, and passive variables of VAEs trained with an increasing number of latent dimensions $|\mathbf{z}|$. (a), (b), and (c) show the results on Symsol, dSprites, and Celeba.

**The IDEs of the model's representations do not change much after the first epoch** The IDEs of the different layers do not change much after the first epoch for well-performing models (see Figure 4). However, for Celeba, whose number of latent dimensions is lower than the data IDE and thus cannot reconstruct the data well, the IDEs tend to change more in the early layers of the encoder, displaying a higher variance.

### 4.3 FINDING THE NUMBER OF DIMENSIONS BY UNSUPERVISED ESTIMATION

As discussed in Section 4.2, the IDEs of the mean and sampled representations start to diverge when (unused) passive variables appear, and this is already visible after the first epochs of training. We can thus use the difference of IDEs between the mean and sampled representations to find the number of latent dimensions retaining the most information while remaining sufficiently compressed (i.e., no passive variables). To this aim, we propose FONDUE: an algorithm to automatically estimate the number latent dimensions for VAEs in an efficient and unsupervised way.

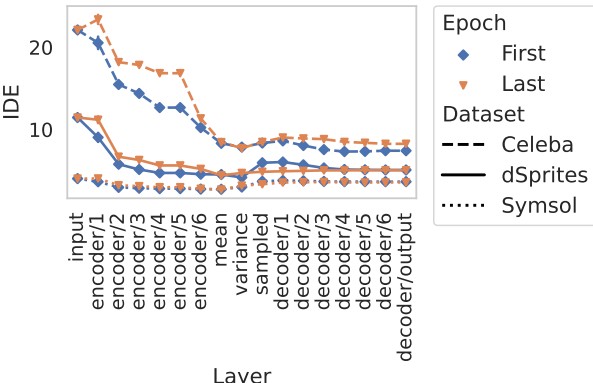

Figure 4: The evolution over multiple epochs of the IDE of the representations learned by VAEs using 10 latent variables on Symsol, dSprites, and Celeba.

**Theorem 1.** *Any execution of FONDUE (Algorithm 1) returns the largest number of dimensions $p$ for which $IDE_z - IDE_\mu \leqslant threshold$, where $IDE_z$, and $IDE_\mu$ are the ID estimates of the sampled and mean representations, respectively.*

*Proof Sketch.* In Algorithm 1, we define a lower and upper bound of the ID estimate, $l$ and $u$, and update the predicted number of latent variables $p$ until, after $i$ iterations, $p_i = l_i$. Using the loop invariant $l_i \leqslant p_i \leqslant u_i$, we can show that the algorithm terminates when $l_i = p_i = \text{floor}\left(\frac{l_i + u_i}{2}\right)$, which can only be reached when $u_i = p_i + 1$, that is, when $p_i$ is the maximum number of latent dimensions for which we have $IDE_z - IDE_\mu \leqslant threshold$. See Appendix A for the full proof. □

**How does FONDUE work?** FONDUE will seek to reach the maximum number of dimensions for which the difference between the mean and sampled IDEs is lower than $t$. At each iteration, it will train a VAE for a few epochs (generally just one) and retrieve the mean and sampled representations corresponding to 10,000 data examples. Using MLE with $k = 20$, we then obtain the (scalar) IDEs of the mean and sampled representations (i.e. $IDE_\mu$ and $IDE_z$) and compute the difference between them. If this difference is lower than or equal to the threshold, we set the current number of latent dimensions to our lower bound and train a VAE again with twice the number of latents, as illustrated in Figure 5. If the difference is higher than the threshold, we set the current number of latent dimensions to our upper bound and train a VAE again with half the number of latents, as illustrated in Figure 6. We iterate these two steps until our current number of latent dimensions is the largest possible dimensionality for which the difference is smaller than or equal to the threshold.

**Obtaining stable estimates** To ensure stable IDEs, we computed FONDUE multiple times, gradually increasing the number of epochs $e$ until the predicted $p$ stopped changing. As reported in Table 1, the results were already stable after one epoch, except for Symsol which needed two[3]. We set a fixed threshold $t = 0.2$ (20% of the data IDE) in all our experiments and used memoisation (see Algorithm 2) to avoid unnecessary retraining and speed up Algorithm 1. For a more in-depth discussion on how to set the threshold value, we refer the reader to Appendix K.

---

[3]Note that the numbers of epochs given in Table 1 correspond to the minimum number of epochs needed for FONDUE to be stable. For example, if we obtain the same score after 1 and 2 epochs, the number of epochs given in Table 1 is 1.

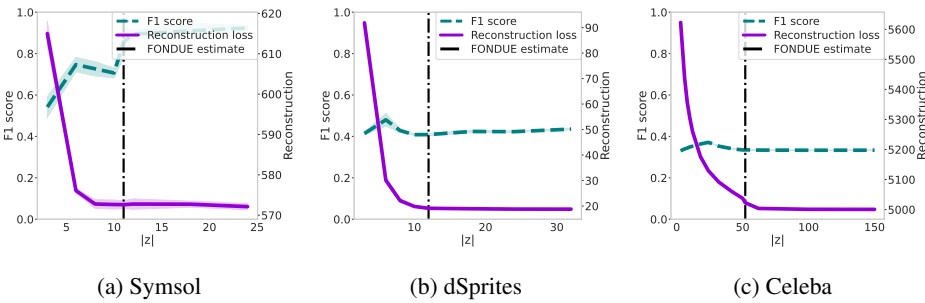

(a) Symsol            (b) dSprites            (c) Celeba

Figure 7: Reconstruction loss and F1 score obtained for generation and downstream tasks of VAEs for Symsol, dSprites, and Celeba with an increasing number of latent variables. The vertical line indicates the number of dimensions found by FONDUE.

Algorithm 1: FONDUE

1: **procedure** FONDUE($t, IDE_{data}, epochs$)
2:     $l \leftarrow 0$                               ▷ Lower bound
3:     $u \leftarrow \infty$                               ▷ Upper bound
4:     $p \leftarrow IDE_{data}$      ▷ Current number of latent dimensions
5:     $mem \leftarrow \{\}$
6:     $threshold \leftarrow t \times p$
7:     **while** $p \neq l$ **do**
8:        $IDE_z, IDE_\mu \leftarrow$ GET-MEM($mem, p, epochs$)
9:        **if** $IDE_z - IDE_\mu \leqslant threshold$ **then**      ▷ Figure 5
10:           $l \leftarrow p$
11:           $p \leftarrow \min(p \times 2, u)$
12:        **else**                         ▷ Figure 6
13:           $u \leftarrow p$
14:           $p \leftarrow$ floor $\left(\frac{l+u}{2}\right)$
15:        **end if**
16:     **end while**
17:     **return** $p$
18: **end procedure**

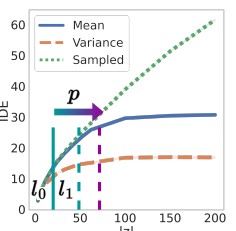

Figure 5: Update $l$ and increase $p$ until $IDE_z - IDE_\mu > threshold$.

Algorithm 2: GET-MEM

1: **procedure** GET-MEM($mem, p, e$)
2:     **if** $mem[p] = \emptyset$ **then**
3:        $vae \leftarrow$ TRAIN-VAE($dim = p, n\_epochs = e$)
4:        $IDE_z, IDE_\mu \leftarrow IDEs(vae)$
5:        $mem[p] \leftarrow IDE_z, IDE_\mu$
6:     **end if**
7:     **return** $mem[p]$
8: **end procedure**

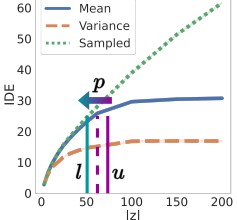

Figure 6: Update $u$ and decrease $p$ until $IDE_z - IDE_\mu \leqslant threshold$.

**Analysing the results of FONDUE** As shown in Table 1, the execution time for finding the number of dimensions for one dataset is much shorter than for fully training one model (approximately 2h using the same GPUs). Moreover, the number of latent dimensions predicted by FONDUE is consistent with existing techniques based on the elbow method (Doersch, 2016; Mai Ngoc & Hwang, 2020) with the additional gain of being obtained in an unsupervised way without fully training any models. Indeed, for all the datasets, the number of dimensions provided by FONDUE corresponds to low reconstruction loss on generation and high F1 scores on downstream tasks.

Table 1: Number of latent variables |**z**| obtained with FONDUE. The results are averaged over 5 seeds, and computation times are reported for NVIDIA A100 GPUs. The computation time is given for one run of FONDUE over the minimum number of epochs needed to obtain a stable score.

| Dataset | Dimensionality (avg $\pm$ SD) | Time/run | Models trained | Epochs/training |
|---------|-------------------------------|----------|----------------|-----------------|
| Symsol  | $11 \pm 0$                     | 7 min    | 6              | 2               |
| dSprites| $12.2 \pm 0.4$                 | 20 min   | 5              | 1               |
| Celeba  | $50.2 \pm 0.9$                 | 14 min   | 9              | 1               |

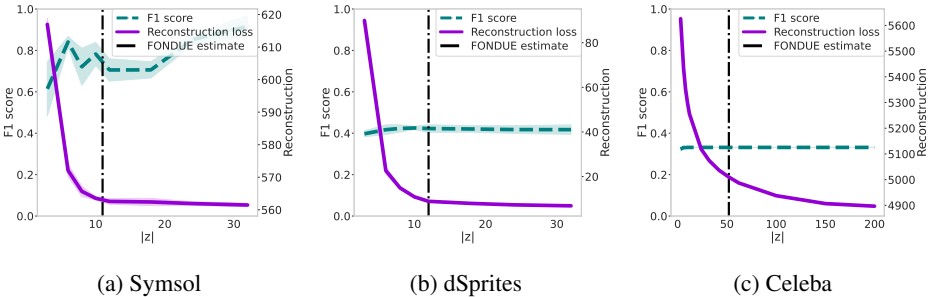

(a) Symsol  (b) dSprites  (c) Celeba

Figure 8: Reconstruction loss and F1 score obtained for generation and downstream tasks of deterministic AEs for Symsol, dSprites, and Celeba with an increasing number of latent variables. The vertical line indicates the number of dimensions found by FONDUE.

**Can FONDUE be applied to other architectures and learning objectives?** We can see in Figure 8 that deterministic AEs with equivalent architectures to the VAEs are performing similarly to VAEs in terms of reconstruction loss using the number of latent dimensions obtained with FONDUE. This is also the case for the F1 score on downstream tasks except for Symsol, which would need more than 30 latent variables to reach an F1 score as good as VAEs. Overall, these results indicate that the dimensionality selected by FONDUE can be reused for AEs trained on the same dataset with an identical architecture for reconstruction, but may provide a lower estimate for downstream tasks in cases where AEs' representations are less efficient than VAEs'. FONDUE also seems to be robust to architectural changes (see Appendix D).

## 5 CONCLUSION

By studying the IDEs of the representations learned by VAEs, we have seen that, very early in the training process, mean and sampled IDEs display increasing discrepancies when the number of latent variable was large enough for passive and mixed variables to appear. These observations lead to FONDUE: an algorithm which can find the number of latent dimensions after which the mean and sampled representations start to strongly diverge. After proving the correctness of our algorithm, we have shown that it is a faster, unsupervised alternative to existing methods which does not require to fully train any model, is not impacted by architectural changes, and can be used for deterministic AEs.

**Future work** While FONDUE has been demonstrated to be an efficient algorithm, it could be improved and extended in several ways: (1) FONDUE is mainly motivated by empirical observation and would benefit from a complementary theoretical approach (e.g., based on well-researched concepts such as information bottleneck (Alemi et al., 2017; Voloshynovskiy et al., 2019)); (2) we have shown that the dimensions given by FONDUE could also be used for deterministic AEs, but it would be interesting to see if this applies to a larger range of unsupervised models (e.g., GANs, clustering methods, etc.); (3) FONDUE can be extended in a number of ways by replacing the difference of IDE in Algorithm 2 by any function that reliably provides different results in mean and sampled representations early in the training. These extensions could be beneficial both in terms of execution time (if the function is faster) and theoretical insights (if the function is more theoretically grounded).

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

## A    PROOF OF THEOREM 1

This section provides the full proof of Theorem 1. To ease its reading, let us first define an axiom based on our observation from Section 4.2 that the IDEs of the mean and sampled representations start to diverge only after the number of latent dimensions has become large enough for (unused) passive variables to appear.

**Axiom 1.** *Let $IDE_x^y$ be the IDE of layer $x$ using $y$ latent dimensions. Given the sets $\mathbb{A} = \{a | IDE_z^a - IDE_\mu^a \leqslant threshold\}$ and $\mathbb{B} = \{b | IDE_z^b - IDE_\mu^b > threshold\}$, we have $a < b \quad \forall\, a \in \mathbb{A},\ \forall\, b \in \mathbb{B}$.*

**Remark 1.** *Given that $l$ and $u$ only take values of latent dimensions for which $IDE_z - IDE_\mu \leqslant threshold$ and $IDE_z - IDE_\mu > threshold$, respectively, Axiom 1 implies that for all iterations $i$, $l_i \in \mathbb{A}$ and $u_i \in \mathbb{B}$ and $l_i < u_i$.*

Using the loop invariant $l_i \leqslant p_i \leqslant u_i$ for each iteration $i$, we will now show that Algorithm 1 terminates when $l_i = p_i = \text{floor}\left(\frac{l_i + u_i}{2}\right)$, which can only be reached when $u_i = p_i + 1$, that is when $p_i$ is the maximum number of latent dimensions for which we have $IDE_z - IDE_\mu \leqslant threshold$.

*Proof.*

**Initialisation:** $l_0 = 0, p_0 = IDE_{data}, u_0 = \infty$, thus $l_0 < p_0 < u_0$.

**Maintenance:** We will consider both branches of the if statement separately:

- For $IDE_z - IDE_\mu \leqslant threshold$ (lines 9-11), $u_i = u_{i-1}$, $p_i = min(p_{i-1} \times 2, u_i)$, and $l_i = p_{i-1}$. We directly see that $p_i \leqslant u_i$. We know from Remark 1 that $l_i < u_i$ and we also have $l_i < p_{i-1} \times 2$, it follows that $l_i < p_i$. Grouping both inequalities, we get $l_i < p_i \leqslant u_i$.

- For $IDE_z - IDE_\mu > threshold$ (lines 12-14), $u_i = p_{i-1}, l_i = l_{i-1}$, and $p_i = \text{floor}\left(\frac{l_i + u_i}{2}\right)$. Using Remark 1 we can directly see that $l_i \leqslant \text{floor}\left(\frac{l_i + u_i}{2}\right) < u_i$ and we obtain $l_i \leqslant p_i < u_i$.

**Termination:** The loop terminates when $l_i = p_i$. Given that $l_i < p_i$ when $IDE_z - IDE_\mu \leqslant threshold$, this is only possible when $IDE_z - IDE_\mu > threshold$, which is when $p_i = \text{floor}\left(\frac{l_i + u_i}{2}\right)$. We know from Remark 1 that $l_i < u_i$, so we must have $(l_i + u_i) \bmod 2 > 0$. As $a \bmod 2 \in \{0, 1\}$, the only possible value for $u_i$ to satisfy $l_i = p_i = \text{floor}\left(\frac{l_i + u_i}{2}\right)$ is $u_i = p_i + 1$. Thus, $p_i$ is the largest number of latent dimensions for which $IDE_z - IDE_\mu \leqslant threshold$.

$\square$

## B    RESOURCES

As mentioned in Sections 1 and 3, we released the code of our experiment, the pre-trained models, and IDEs:

- The IDEs can be downloaded from an anonymous Google account using the following tiny URL `https://t.ly/8r3N`

- `symsol_reduced`, the reduced version of Symmetric solids, can be downloaded using an anonymous Google account using the following tiny URL `https://t.ly/_CdH`

- The code can also be downloaded from an anonymous Google account using another tiny URL `t.ly/Oh7s`

- Our pre-trained models are large and could not be shared with the reviewers using an anonymous link. The URL to the models will, however, be available in the non-anonymised version of this paper.

The 300 models correspond to 5 runs of VAEs trained with:

- 8 choices of latent dimensions for Symsol and dSprites, using convolutional and fully-connected architectures, resulting in 160 models
- 14 choices of latent dimensions for Celeba, using convolutional and fully-connected architectures, resulting in 140 models

The total 300 pre-trained models were then used to compute estimate IDs as described below. Note that while these models would save some computational time if used to reproduce the experiment, they are only provided to reduce the carbon footprint of reproducing the experiment as one could easily retrain the models using the details of our implementation.

The IDEs mentioned above correspond to the 35,000 ID estimates of the 14 layers of the 5 runs of the VAEs considered, and computed using 5 configurations (MLE with $k = 3, 5, 10, 20$ and TwoNN) where the MLE was computed 3 times for each $k$, using different seeds. We computed these for:

- 8 choices of latent dimensions for Symsol and dSprites, resulting in 16,800 ID estimates.
- 14 choices of latent dimensions for Celeba, resulting in 14,700 ID estimates.
- 1 choice of latent dimension with $\beta = 20$ for dSprites, resulting in 1,050 ID estimates.
- 1 choice of latent dimension at 1 early epoch for dSprites, Celeba and Symsol, resulting in 3,150 ID estimates.

The total 35,700 IDEs were then used in Figures 2, 4, and 13.

## C  EXPERIMENTAL SETUP

Our implementation uses the same hyperparameters as Locatello et al. (2019), as listed in Table 2. We reimplemented the Locatello et al. (2019) code base, designed for Tensorflow 1, in Tensorflow 2 using Keras. The model architectures used are also similar, as described in Table 3 and 4. We used the convolutional architecture in the main paper and the fully-connected architecture in Appendix D. Each model is trained 5 times with seed values from 0 to 4. Every image input is normalised to have pixel values between 0 and 1. TwoNN is used with an anchor of 0.9, and the hyperparameters for MLE can be found in Table 5.

Table 2: VAEs hyperparameters

| Parameter | Value |
|---|---|
| Batch size | 64 |
| Latent space dimension | 3, 6, 8, 10, 12, 18, 24, 32. |
| | For Celeba only: 42, 52, 62, 100, 150, 200 |
| Optimizer | Adam |
| Adam: $\beta_1$ | 0.9 |
| Adam: $\beta_2$ | 0.999 |
| Adam: $\epsilon$ | 1e-8 |
| Adam: learning rate | 0.0001 |
| Reconstruction loss | Bernoulli |
| Training steps | 300,000 |
| Train/test split | 90/10 |
| $\beta$ | 1 |

## D  FONDUE ON FULLY-CONNECTED ARCHITECTURES

We report the results obtained by FONDUE for fully-connected (FC) architectures in Table 6 and Figure 9. As shown in Table 6, the execution time for estimating the number of dimensions for one dataset is much shorter than for training one model (this is approximately 2h on the same GPUs), in similarity with convolutional VAEs. As in Section 4.3, FONDUE correctly finds the number of latent dimensions after which the mean and sampled IDEs start to diverge, as shown in Figure 9. One can see that the number of latent variables needed for FC VAEs is much lower than

Table 3: Architecture

| Encoder | Decoder |
|---|---|
| Input: $\mathbb{R}^{64 \times 63 \times channels}$ | $\mathbb{R}^{10}$ |
| Conv, kernel=4×4, filters=32, activation=ReLU, strides=2 | FC, output shape=256, activation=ReLU |
| Conv, kernel=4×4, filters=32, activation=ReLU, strides=2 | FC, output shape=4x4x64, activation=ReLU |
| Conv, kernel=4×4, filters=64, activation=ReLU, strides=2 | Deconv, kernel=4×4, filters=64, activation=ReLU, strides=2 |
| Conv, kernel=4×4, filters=64, activation=ReLU, strides=2 | Deconv, kernel=4×4, filters=32, activation=ReLU, strides=2 |
| FC, output shape=256, activation=ReLU, strides=2 | Deconv, kernel=4×4, filters=32, activation=ReLU, strides=2 |
| FC, output shape=2x10 | Deconv, kernel=4×4, filters=channels, activation=ReLU, strides=2 |

Table 4: Fully-connected architecture

| Encoder | Decoder |
|---|---|
| Input: $\mathbb{R}^{64 \times 63 \times channels}$ | $\mathbb{R}^{10}$ |
| FC, output shape=1200, activation=ReLU | FC, output shape=256, activation=tanh |
| FC, output shape=1200, activation=ReLU | FC, output shape=1200, activation=tanh |
| FC, output shape=2x10 | FC, output shape=1200, activation=tanh |

Table 5: MLE hyperparameters

| Parameter | Value |
|---|---|
| k | 3, 5, 10, 20 |
| anchor | 0.8 |
| seed | 0 |
| runs | 5 |

for convolutional VAEs (see Table 1 for comparison). For dSprites, it is near the true ID of the data, and for Celeba, it is close to the data IDE reported in Figure 1 of Section 4.1.

As in Section 4.3, we gradually increase the number of epochs until FONDUE reaches a stable estimation of the latent dimensions. As these models have fewer parameters than the convolutional architecture used in Section 4.3, they converge more slowly and need to be trained for more epochs on Celeba and Symsol before reaching a stable estimation (Arora et al., 2018; Sankararaman et al., 2020). dSprites contains more data examples than the other datasets and less complex data than Celeba, which can explain its quicker convergence.

For dSprites and Symsol, the number of dimensions selected by FONDUE corresponds to the number of dimensions after which the reconstruction stops improving and the KL divergence remains stable (see Figure 10). For Celeba, the reconstruction continues to improve slightly after 39 latent dimensions, due to the addition of variables between 42 and 100 latent dimensions, as illustrated in Figure 11. As in convolutional architectures, one could increase the threshold of FONDUE to take more mixed variables into account.

Overall, we can see that FONDUE also provides good results on the FC architectures, despite a slower convergence, showing robustness to architectural changes.

Table 6: Number of latent variables obtained with FONDUE for fully-connected architectures. The results are averaged over 5 seeds, and computation times are reported for NVIDIA A100 GPUs. The computation time is given for one run of FONDUE over the minimum number of epochs needed to obtain a stable score.

| Dataset | Dimensionality (avg $\pm$ SD) | Time/run | Models trained | Epochs/training |
|---------|------------------------------|----------|----------------|-----------------|
| Symsol  | $8 \pm 0$                     | 15 min   | 6              | 6               |
| dSprites| $6.6 \pm 0.5$                 | 16 min   | 5              | 1               |
| Celeba  | $39 \pm 0.6$                  | 50 min   | 7              | 9               |

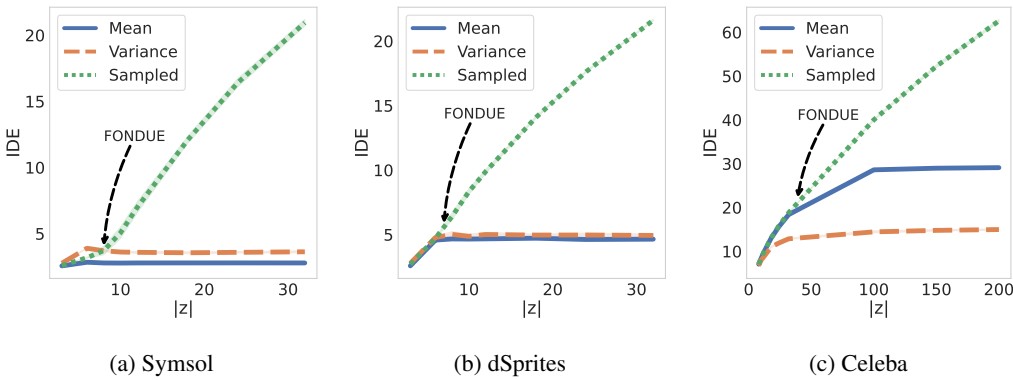

(a) Symsol       (b) dSprites       (c) Celeba

Figure 9: Number of latent dimensions provided by FONDUE for fully-connected VAEs: $|\mathbf{z}| = 8$ on Symsol, $|\mathbf{z}| = 7$ on dSprites, and $|\mathbf{z}| = 39$ on Celeba.

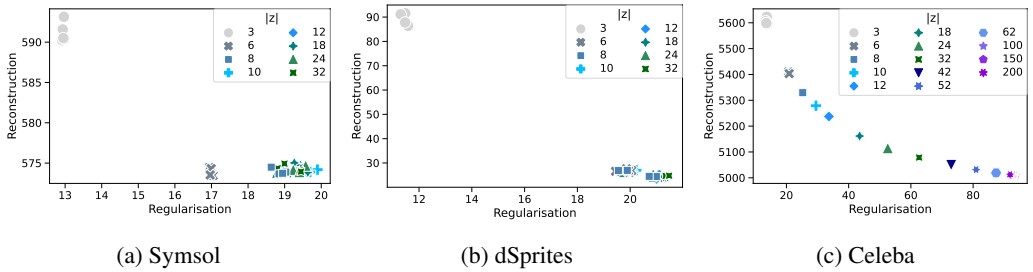

(a) Symsol       (b) dSprites       (c) Celeba

Figure 10: Reconstruction and KL divergences (i.e., regularisation scores) of fully-connected VAEs for Symsol, dSprites, and Celeba with an increasing number of latent variables $|\mathbf{z}|$.

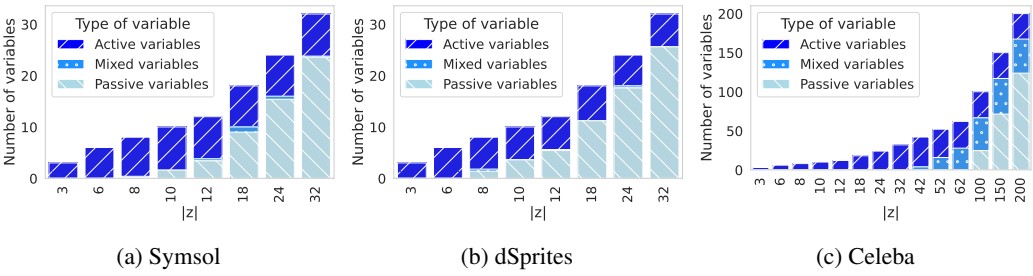

(a) Symsol       (b) dSprites       (c) Celeba

Figure 11: Quantity of active, mixed, and passive variables of VAEs trained with an increasing number of latent dimensions $|\mathbf{z}|$. (a), (b), and (c) show the results on Symsol, dSprites, and Celeba, respectively.

# E  ADDITIONAL DETAILS ON MEAN, VARIANCE, AND SAMPLED REPRESENTATIONS

This section presents a concise illustration of what mean, variance, and sampled representations are. As shown in Figure 12, the mean, variance and sampled representations are the last 3 layers of the encoder, where the sampled representation, **z**, is the input of the decoder.

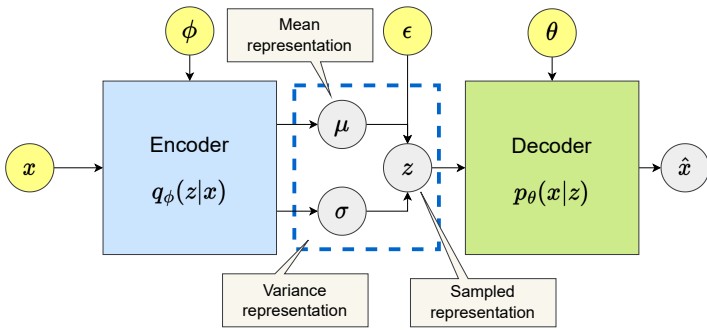

Figure 12: The structure of a VAE

# F  PASSIVE VARIABLES AND POSTERIOR COLLAPSE

As discussed in Section 2.1, passive variables appear in latent representations of VAEs in two cases: polarised regime and posterior collapse. In well-behaved VAEs (i.e., in the case of polarised regime), passive variables arise when the number of latent dimensions is larger than the number of latent variables needed by the VAE to encode latent representations. These passive variables contribute to lowering the regularisation loss term of Equation 1 without increasing the reconstruction loss. However, passive variables can also be encountered as part of a pathological state where the reconstruction loss is very high and the regularisation loss is pushed towards zero (i.e., when posterior collapse takes place). This issue can happen for various reasons (Dai et al., 2020), but is clearly distinct from the polarised regime as the reconstruction loss is very high and the latent representations contain little to no active variables.

In both cases, passive variables are very different in mean and sampled representations, due to the sampling process $\mathbf{z} \sim \boldsymbol{\mu} + \epsilon\sqrt{\sigma\boldsymbol{I}}$, where $\epsilon \sim \mathcal{N}(0, \boldsymbol{I})$, $\boldsymbol{\mu}$ is the mean representation and $\sigma\boldsymbol{I}$ the diagonal matrix of the variance representations. For the regularisation term to be low, one needs to create passive variables, that is, as many dimensions of $\mathbf{z}$ as close as possible to $\mathcal{N}(0, \boldsymbol{I})$. This can easily be done by setting some elements of $\boldsymbol{\mu}$ to 0 and their corresponding variance to 1. As a result, passive variables, when observed over multiple data examples will have a mean of 0 in the mean and sampled representations. However, their variance will be close to 0 in the mean representations, and close to 1 in their sampled counterpart (Rolinek et al., 2019; Bonheme & Grzes, 2021).

# G  WHY NOT USE VARIABLE TYPE INSTEAD OF IDE FOR FONDUE?

As passive variables are easy to detect, one could wonder why they were not used directly to determine the number of latent dimensions instead of comparing IDEs of models trained for a few epochs multiple times. For example, it would be quicker to train one model with a large number of latent variables for a few epochs and retrieve the number of active (or active and mixed) variables detected, as for example, illustrated in Algorithm 3.

**How does Algorithm 3 work?**   We define the initial number of latent variables as twice the data IDE. Then, if we want to have enough dimensions for active and mixed variables, we double the number of latent variables until we find at least one passive variable and return the sum of active and mixed variables as the chosen number of latent dimensions. If we want instead to have only active variables, we double the number of latent variables until we find at least either one passive or mixed variable and return the number of active variables as the chosen number of latent dimensions.

**Why use Algorithm 1 instead?**    As shown in Table 7, the identification of variable types displays a high variance during early training, which generally makes Algorithm 3 less reliable than Algorithm 1 for equivalent computation time. In addition to this instability, the numbers of latent dimensions predicted by Algorithm 3 in Table 7 are far from optimal compared to Table 1. There is a large overestimation in Symsol and an underestimation in Celeba. These issues may be explained by the fact that Algorithm 1 is based on mean and sampled representations, while Algorithm 3 solely relies on variance representations, decreasing the stability during early training. Moreover, mixed and passive variables are not discriminated correctly in early epochs, possibly for the same reasons, preventing any modulation of compression/reconstruction quality that could be achieved with Algorithm 1.

---

**Algorithm 3** FONDUE with variable types

---

**procedure** FONDUE-VAR($data\_ide, epochs, keep\_mixed$)
    $l \leftarrow 2 * data\_ide$
    $n \leftarrow -1$
    **while** $n < 0$ **do**
        $vae \leftarrow train\_VAE(dim = l, n\_epochs = epochs)$
        $av, mv, pv \leftarrow variable\_types(vae)$    ▷ Number of active, mixed and passive variables
        **if** $pv > 0$ and $keep\_mixed$ **then**
            $n \leftarrow av + mv$
        **else if** $(mv > 0$ or $pv > 0)$ and not $keep\_mixed$ **then**
            $n \leftarrow av$
        **else**
            $l \leftarrow l * 2$
        **end if**
    **end while**
    **return** $n$
**end procedure**

---

Table 7: Number of latent variables obtained with FONDUE-var. The results are averaged over 5 seeds and computation times are reported for NVIDIA A100 GPUs.

| Dataset | Dimensionality (avg $\pm$ SD) | Time/run | Models trained | Epochs/training |
|---------|-------------------------------|----------|----------------|-----------------|
| Symsol  | $14 \pm 1.2$   | 2 min  | 3 | 1 |
| Symsol  | $17 \pm 1.7$   | 3 min  | 3 | 2 |
| Symsol  | $15 \pm 1.1$   | 10 min | 3 | 5 |
| dSprites | $9.2 \pm 1.3$ | 4 min  | 1 | 1 |
| dSprites | $8.8 \pm 1.2$ | 8 min  | 1 | 2 |
| dSprites | $9.6 \pm 0.5$ | 20 min | 1 | 5 |
| Celeba  | $38.6 \pm 2.6$ | 1 min  | 1 | 1 |
| Celeba  | $38.6 \pm 0.4$ | 2 min  | 1 | 2 |
| Celeba  | $41.2 \pm 0.4$ | 6 min  | 1 | 5 |

## H  ADDITIONAL DETAILS ON ID ESTIMATION USING MLE

The objective of this section is to provide an intuitive view of ID estimation using MLE. We refer the reader to Section 2.2 and Levina & Bickel (2004) for a more technical discussion.

Levina & Bickel (2004) model the number of neighbours of a point $\boldsymbol{X}_i$ in a radius $R$ using a Poisson process. This Poisson process, $\{N(t, \boldsymbol{X}_i), 0 \leqslant t \leqslant R\}$, will count the total number of points falling into the successive $d$-dimensional spheres of radius $0 \leqslant t \leqslant R$. Intuitively, when $d = 3$ this can be thought of as an onion to which we add an outer peel for each increasing radius value $t$, until we reach the maximum radius $R$. Thus, each $N(t, \boldsymbol{X}_i)$ will give us a snapshot of the number of points contained in all the peels stacked so far in the onion of radius $t$.

Table 8: Number of latent variables $|\mathbf{z}|$ obtained with FONDUE using MLE with different number of neighbours $k$. The results are averaged over 5 seeds, and computation times are reported for NVIDIA A100 GPUs. We retain the same number of epochs as in Table 1.

| k | $|z|$ (avg $\pm$ SD) Symsol | $|z|$ (avg $\pm$ SD) dSprites | $|z|$ (avg $\pm$ SD) Celeba |
|----|----|----|----|
| 3 | 1.6±1.2 | 9.4±0.5 | 45.2±0.6 |
| 5 | 4.4±1.4 | 10.4±0.8 | 46.8±1.1 |
| 10 | 5±2 | 10.4±0.5 | 48.8±1.0 |
| 20 | 11 ± 0 | 12.2 ± 0.4 | 50.2 ± 0.9 |

As $N(t, \boldsymbol{X}_i)$ is a function of the surface area of the sphere, its rate is a function of $d$ and one can estimate $d$ using MLE. However, we generally cannot access all the existing neighbours of $\boldsymbol{X}_i$ in a given radius without infinite data, so we approximate the process using a fixed number of neighbours.

Now, let us consider the point $\boldsymbol{X}_i = (0, 0, 0)$ and 3 closest neighbours $\boldsymbol{Y} = \{(0, 1, 0)(1, 0, 0), (2, 0, 0)\}$. We have $N(t = 1, X_i) = 2$ and $N(t = 2, X_i) = 3$ because $\boldsymbol{Y}_1, \boldsymbol{Y}_2$ are within a radius $t = 1$ of $\boldsymbol{X}_i$, and all $\boldsymbol{Y}_j$ are within a radius $t = 2$.

Using the distances between $\boldsymbol{X}_i$ and its neighbours, $T(\boldsymbol{X}_i, \boldsymbol{Y}_j)$, the dimensionality can be estimated by Equation 4 as follows.

$$
\begin{aligned}
\bar{d}_3(\boldsymbol{X}_i) &= \left[ \frac{1}{2} \sum_{j=1}^{2} \log \frac{T(\boldsymbol{X}_i, \boldsymbol{Y}_3)}{T(\boldsymbol{X}_i, \boldsymbol{Y}_j)} \right]^{-1}, \\
&= \left[ \frac{1}{2} \sum_{j=1}^{2} \log \frac{2}{T(\boldsymbol{X}_i, \boldsymbol{Y}_j)} \right]^{-1}, \\
&= [\log 2]^{-1}, \\
&\approx 3.3,
\end{aligned}
\tag{8}
$$

which is reasonably close to the true data ID.

To make sure that the estimate is stable, we repeat this estimation over multiple data points and average the results as per Equation 5.

# I    IMPACT OF $k$ ON FONDUE

MLE is sensitive to the number of neighbours $k$, and to the best of our knowledge, there is no principled method to select it. As mentioned in Section 4.1, our objective is to compare the IDEs of different representations, not to accurately estimate their true ID. Thus, having stable ID estimates is as important as than being as close as possible to the true ID, which is what guided our selection process for $k$. We selected $k$ based on a combination of multiple criteria to ensure consistency. Specifically, we ensured that:

- the amount of data used was sufficiently large (10,000 samples) and the selected $k$ provided a stable estimate over multiple runs, as recommended by Levina & Bickel (2004),

- the IDEs obtained with the selected $k$ were similar to the IDEs obtained with the closest value of $k$, as recommended by (Karbauskaitė et al., 2011),

- the IDEs obtained with the selected $k$ were generally similar to the estimated values provided by TwoNN.

As shown in Table 8, the results obtained by FONDUE with $k < 20$ are generally either less consistent (e.g. see large variance of estimates for Symsol), or close to the estimates obtained with $k = 20$ (e.g., see dSprites). Moreover, the dimensionality obtained with $k = 20$ aligns best with good reconstruction scores, and equivalent or better performance on downstream tasks, as seen in Figure 7.

## J    NON-VARIATIONAL AUTOENCODERS

Deep deterministic autoencoders (AEs) (Kramer, 1991) can be thought of as a non-linear version of PCA (Pearson, 1901). They are composed of an encoder $f_\phi(\boldsymbol{X})$ which maps the data points $\boldsymbol{X}$ to compressed representations $\boldsymbol{Z}$, and a decoder $g_\theta(\boldsymbol{Z})$ which attempt to reconstruct $\boldsymbol{X}$ from compressed representation $\boldsymbol{Z}$. AEs are optimised to minimise the reconstruction error $\mathcal{L}(\boldsymbol{X}, g_\theta(\boldsymbol{Z}))$ (e.g., MSE).

## K    HOW TO SELECT A GOOD VALUE OF $t$?

While $t$ was set to a fixed value of 0.2 in this paper's experiment, one could wonder if this would be a good fit for their particular use case, and, if not, how to choose the value of $t$. While we do not have an automated way to select this parameter, we believe that having an intuitive idea of what $t$ represents could inform such a decision.

By looking at Figures 2 and 3, one can see that the difference between the IDEs of the mean and sampled representations is generally close to the number of additional mixed and passive variables. $t$ represents this number of "extra variables" (mixed and passive) that we want to allow the model to use. As more complex datasets with high ID generally display more mixed variables, we set $t$ as a fraction of the data IDE to allow the number of "extra" variables selected to scale with the IDE. For example, given data with IDE of 25, $t = 0.2$ would roughly be equivalent to allowing for $25 * 0.2 = 5$ non-active variables (mixed or passives) in the representation chosen.

Allowing more "extra" variables (larger $t$) will generally provide higher likelihood and could thus be considered when the focus is on data generation of complex datasets (where these types of variables can generally be learned). However, too many extra variables may decrease the accuracy on downstream tasks (Mai Ngoc & Hwang, 2020) and lead to an increased correlation of the mean representations (Bonheme & Grzes, 2021). So, the value of $t$ will mainly depend on complexity of the dataset and the goal of the model, which is why we cannot provide a principled way to define it. However, our experiment shows that $t = 0.2$ is a good initial value.

## L    ADDITIONAL RESULTS

This section provides additional observations of the IDEs of VAEs which are complementary to section 4.2 but not necessary for understanding FONDUE.

**What happens in the case of posterior collapse?**   By using a $\beta$-VAE with very large $\beta$ (e.g., $\beta = 20$), one can induce posterior collapse, where a majority of the latent variables become passive and prevent the decoder from accessing sufficient information about the input to provide a good reconstruction. This phenomenon is illustrated in Figure 13d, where the IDEs of the encoder representations are similar to what one would obtain for a well performing model in the first 5 layers, indicating that these early layers of the encoder still encode some useful information about the data. The IDEs then drop in the last three layers of the encoder, indicating that most variables are passive, and only a very small amount of information is retained. The IDE of the sampled representation (see *sampled* in Figure 13d) is then artificially inflated by the passive variables and becomes very close to the number of dimensions $|\mathbf{z}|$. From this, the decoder is unable to learn much and has thus a low IDE, close to the IDE of the mean representation (see the points on the RHS of Figure 13d).

**The IDEs of the encoder representations decrease, but the IDEs of the decoder representations stay constant**   We can see in Figure 13 that the IDE of the representations learned by the encoder decreases until we reach the mean and variance layers, which is consistent with the observations reported for classification (Ansuini et al., 2019). Interestingly, for dSprites and Symsol, when the number of latent variables is at least equal to the IDE of the data, the IDE of the mean and variance representations is very close to the true data ID. After a local increase of the IDE in the sampled representations, the IDE of the decoder representations stays close to the IDE of the mean representations and does not change much between layers.

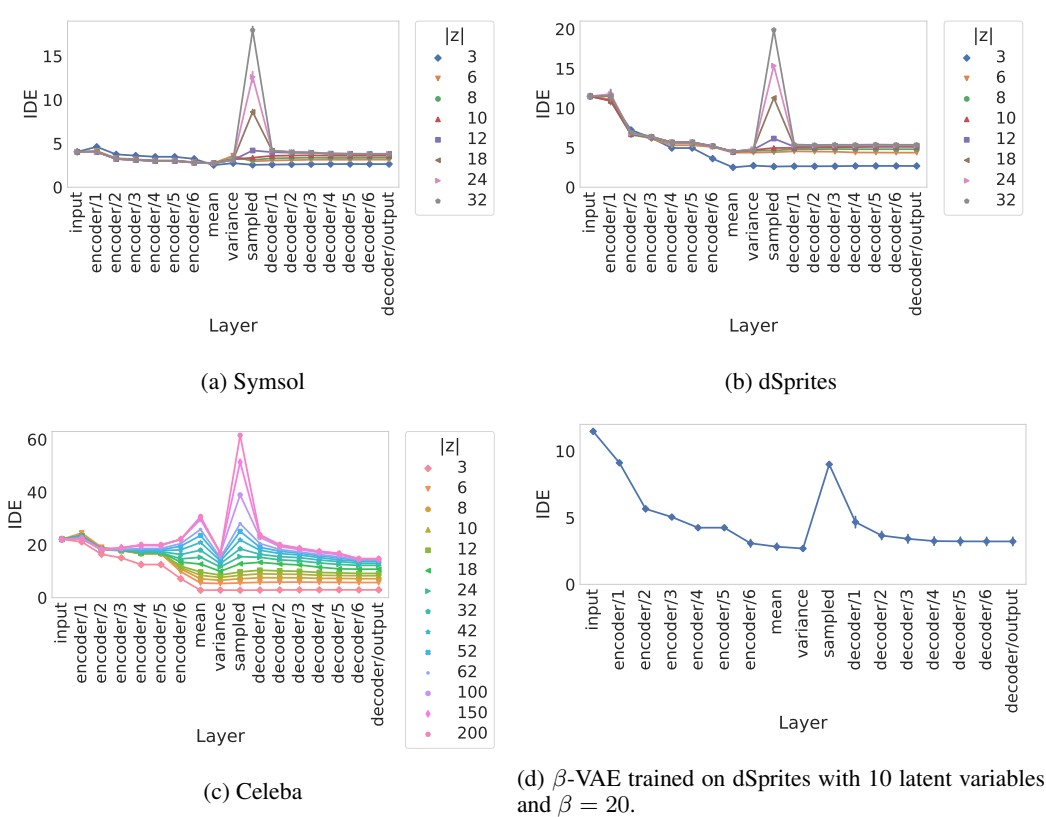

Figure 13: IDEs of VAEs trained with an increasing number of latent dimensions $|\mathbf{z}|$. (a), (b), and (c) show the results on Symsol, dSprites, and Celeba, respectively. (d) shows the results of $\beta$-VAEs trained on dSprites with 10 latent variables and $\beta = 20$ to cause posterior collapse.

