# OpenReview forum: "FONDUE: an Algorithm to Find the Optimal Dimensionality of the Latent Representations of Variational Autoencoders"
_ICLR.cc/2023/Conference — Submitted to ICLR 2023_

### Official Review · Reviewer_EweH · 2022-10-18

**Confidence:** 5
**Correctness:** 2
**Technical Novelty And Significance:** 2
**Empirical Novelty And Significance:** 2
**Recommendation:** 3

**Clarity, Quality, Novelty And Reproducibility:**

The article reads very well, it covers the basics with a useful background section, and presents both the experimental protocol and proposed method clearly.
As noted in the previous part of the review, I am surprised to see no related work at all. This also hurts the experimental evaluation of FONDUE as it is not compared to existing alternatives (albeit Appendix G is an effort in the right direction).
Concerning the novelty, this is somehow hard to judge, since no related work is discussed, and since the method borrows known methods as the key enabler for the proposed approach. The FONDUE algorithm per-se is simple (which is not used here as a bad connotation), but it is based on heuristics and approximations (because the methods it relies on are approximations of the ID). A possible suggestion to further develop theoretically grounded bases for this work is to dig into the information-theory spin that several work has proposed to study VAEs, such that those based on the information bottleneck principle, see e.g. [1, 2].

[1] @inproceedings{
alemi2017deep,
title={Deep Variational Information Bottleneck},
author={Alexander A. Alemi and Ian Fischer and Joshua V. Dillon and Kevin Murphy},
booktitle={International Conference on Learning Representations},
year={2017},
url={https://openreview.net/forum?id=HyxQzBceg}
}

[2] @misc{https://doi.org/10.48550/arxiv.1912.00830,
  url = {https://arxiv.org/abs/1912.00830},
  author = {Voloshynovskiy, Slava and Kondah, Mouad and Rezaeifar, Shideh and Taran, Olga and Holotyak, Taras and Rezende, Danilo Jimenez},
  title = {Information bottleneck through variational glasses},
  publisher = {arXiv},
  year = {2019},
}


**Strength And Weaknesses:**

The strength of this paper are:
* This work is very clear and well motivated
* The proposed approach is simple and relies on thoughtful observations from an empirical analysis of the virtues of existing IDE methods


The weaknesses of this paper are:
* I am missing an appropriate evaluation of the impact of an informed choice of the latent space dimension on performance, such as a table with likelihood values. Fig 8 and Fig 9 are difficult to digest, whereas a simple table with likelihood values for different $|z|$ would be perfect.
* Claims about optimality overstate the contribution. There is no proof of optimality in this work, so I suggest the authors to either rephrase their statements, or to try and prove that the output of the FONDUE algorithm is optimal. For example, synthetic datasets are endowed with ground truth number of factors of variation; the value for the latent dimension is larger than the ground truth (due to over-estimates caused by IDE methods used in this work), which implies that the found solution is not optimal
* The threshold $t$ is set arbitrarily and heuristically. Is there any informed principle to pick a good $t$?
* One additional observation is as follows. In practical endeavors, it is frequent to see VAEs or AEs to be used as a “pre-training” step, with a reconstruction objective. Then, decoders are disconnected, encoders are freezed, and the latent representation of the input is used as input to a task-specific architecture. Is the “ideal” latent size obtained by FONDUE still relevant, for example, in terms of downstream performance?
* Apart from Appendix G, which is commendable as it presents an alternative to FONDUE, I do not see any alternative methods, nor a related work section. Is it possible nowadays that nobody else in the huge body of work that exists on VAEs, has explored the same research question addressed in this work?


**Summary Of The Paper:**

The work presented in this paper is of empirical nature, and attempts at determining the number of latent variables to be used in generic Variational Autoencoder (VAE) models. This is done by relying on two existing methods to estimate the intrinsic dimension (ID) of i.i.d. input samples, that use the concept of neighborhood to approximate the data intrinsic dimension.

The goal of the experimental setup is to study the intrinsic dimensions of the representations learned by VAEs to assess whether they can be used to determine the optimal number of latent dimensions of VAEs. More precisely, the proposed experimental methodology is to compute the ID for all layers’ activations of several variants of a VAE model, each with an increasing dimension of the latent space.
The ultimate goal of this empirical study is to design a new algorithm to automatically find the optimal number of latent dimensions for VAEs in an efficient and unsupervised way.

As for the results of the study, the authors first attempt at validating the two methods to estimate the data ID, which indicates that the used methods tend to overestimate the ID.

Then, the authors move to applying ID estimation to intermediate layers of a VAE, trained with different latent dimensions, and present a number of findings, that can be summarized as: if the dimension of the latent space is correctly defined (such that it is at least equal to the ID of the input data), then the behavior of the ID at each layer of the encoder/decoder architecture follows a well defined pattern, the ID of the latent space corresponds to the data ID, whereas the sampled distributions usually having a much larger ID.

Based on these findings, the authors present the algorithm called FONDUE, which builds on the key observation that the IDs of the mean and sampled representations start to diverge (from the ID of the data distribution) when (unused) passive variables appear, and this is already visible after the first epochs of training. After proving the correctness of the FONDUE algorithm in terms of its termination, a series of experiments show that the algorithm is a computationally efficient alternative to grid search to find an appropriate latent space size.


**Summary Of The Review:**

This paper presents a clear empirical study and a simple heuristic to determine a good size for the latent dimension of a variational autoencoder. My main concerns relate to 1) claims of optimality, 2) lack of comparison with alternatives, 3) a somewhat limited study in terms of practical use of VAEs for learning representations that are useful for downstream tasks.

---

> ### Author Response · Authors · 2022-11-11
> **Answer to reviewer EweH**
>
> Thank you for your interest in our work and for your detailed comments.
>
> Answers to Reviewer's Questions:
> --------------------------------
>
> >  I am missing an appropriate evaluation [...] would be perfect.
>
> We have updated Fig 8 and 9 to make them clearer but choose not to replace them by a table.
> Indeed, current methods [1,3] (see related question below) select the number of dimensions based on the reconstruction loss or downstream task score.
> Specifically this is done using the elbow method (i.e., choosing the dimensionality corresponding to where a curve "bends") to select
> the number of dimensions based on the reconstruction loss or downstream task score.
> Thus, to relate FONDUE to [1,3], we thought it was important to illustrate where the dimensionality selected was on those curves.
> To improve the readability, Fig 8 and 9 are now line plots with $|z|$ in the x-axis and the reconstruction loss and downstream tasks F1 score as two separate y-axis.
> The dimensionality selected by FONDUE is indicated by a vertical line, making clear how good the reconstruction loss and quality on downstream task of a model trained
> with this number of latent dimensions will be.
>
> > Claims about optimality [...] not optimal
>
> We apologise for the misunderstanding, our intent was not to claim "optimal" results in the sense of the number of
> dimensions provided by FONDUE matching the data ID (This would lead to lower performance of VAEs as shown in Fig. 8 and 9) but
> "optimal" in the sense that given a threshold $t$, FONDUE is guaranteed to provide the largest possible number of dimensions
> for which the difference between the mean and sampled IDEs is smaller than $t$ as shown in Appendix A.
> However, we agree that this could be confusing and have now removed any mention of "optimal dimensions".
>
> >  The threshold $t$ [...] pick a good $t$?
>
> By looking at Fig. 3 and 4, one can see that the difference between the IDEs of the mean and sampled
> representation is generally close to the number of additional mixed and passive variables.
> $t$ represents this number of "extra variables" (mixed and passive) that we want to allow the model to use.
> As more complex datasets with high IDE generally display more mixed variables, we set $t$ as a fraction of the data IDE
> to allow the number of "extra" variables selected to scale with the IDE.
> For example, given data with IDE of 25, $t=0.2$ would roughly be equivalent to allowing for $25 * 0.2 = 5$ non-active variables (mixed or passives) in the representation chosen.
> However, too many extra variables may decrease the accuracy on downstream tasks [1] and lead to an increased correlation of the mean representations [2].
> So, the value of $t$ will mainly depend on the complexity of the dataset and the goal of the model, which is why we cannot provide a principled way to define it.
> However, we believe that the approximate relationship between $t$ and the number of extra variables allowed could be useful in informing this choice.
> As this question may be important for many practitioners, we have added a slightly updated version of this discussion in appendix K to explain what different values of $t$ will do.
>
> > One additional observation [...] downstream performance?
>
> Thank you for your suggestion. We now present the F1 score of models with different latent dimensions on classification tasks in Sec. 4.
> This also allows us to compare the dimensions obtained by FONDUE with the results of [1] who selected the number of latent dimensions based on the performance on downstream tasks (see next answer).
>
> > Apart from Appendix G [...] addressed in this work?
>
> This question seems to have been scarcely discussed and exclusively in the context of post-hoc comparisons of fully trained models.
> We have now added [1,3] in the related work section as they propose some criteria to select the best number of latent dimensions
> when comparing multiple fully trained models. As explained above, these studies are based on the elbow method and require human supervision to
> select the number of latent dimensions. However, they were the closest contributions we could find despite a throughout search for related work.
> Additionally, we now discuss how FONDUE meets the criteria set by [1,3]] in section 4.
> In an attempt to encourage more work on this question, in Sec. 5, we also discuss possible extensions of FONDUE that could be investigated.
>
> References:
> -----------
> - [1] Kien Mai Ngoc and Myunggwon Hwang. Finding the best k for the dimension of the latent space in autoencoders. In Computational Collective Intelligence, pp. 453–464. Springer International Publishing, 2020. ISBN 978-3-030-63007-2.
> - [2] Lisa Bonheme and Marek Grzes. Be More Active! Understanding the Differences between Mean and Sampled Representations of Variational Autoencoders. arXiv e-prints, 2021.
> - [3] Carl Doersch. Tutorial on Variational Autoencoders. arXiv e-prints, 2016.
>
> (comments continue on next page)

---

> > ### Author Response · Authors · 2022-11-11
> > **Answer to reviewer EweH (part 2)**
> >
> > Other comments:
> > ---------------
> > > A possible suggestion [...] information bottleneck principle, see e.g. [1, 2].
> >
> > We agree that it would be interesting to see extensions of FONDUE which are more theoretically grounded, and IB could be
> > a promising way to do so. We have now mentioned this in the future work part of the conclusion.
> >
> > > My main concerns relate to [...] downstream tasks.
> >
> > Your interest in our work is very much appreciated.
> > We hope that our reformulation regarding optimality, the addition of related work and evaluation on downstream tasks will address your concerns.
> >
> > References:
> > -----------
> > - [1] Kien Mai Ngoc and Myunggwon Hwang. Finding the best k for the dimension of the latent space in autoencoders. In Computational Collective Intelligence, pp. 453–464. Springer International Publishing, 2020. ISBN 978-3-030-63007-2.
> > - [2] Lisa Bonheme and Marek Grzes. Be More Active! Understanding the Differences between Mean and Sampled Representations of Variational Autoencoders. arXiv e-prints, 2021.
> > - [3] Carl Doersch. Tutorial on Variational Autoencoders. arXiv e-prints, 2016.

---

> > > ### Comment · Reviewer_EweH · 2022-11-21
> > > **Thank you for your detailed answers**
> > >
> > > I would like to thank the authors for their detailed answers to my questions. I have read all other reviews,  comments, and authors’ answers, as well as read carefully the updated paper.
> > > Despite I think the authors did a very good job in addressing the various questions, run several additional experiments, clarified both in the main paper and the appendix the parts that required more explanations, I am still left with questions.
> > >
> > > 1. Experiments with a simple downstream task. The findings are interesting and corroborate some previous work cited by the authors. For downstream performance, a large number of latent variables is not necessarily a good option, as shown in the results: F1-scores are sometimes higher for lower latent dimensionality, smaller than that found by FONDUE. Besides some peaks, F1-scores seem also not very affected by the latent dimensionality (dSprites, CelebA), so whatever we chose as $|z|$ seem to work.
> > >
> > > Now, the authors indicate that their main purpose is to find a $|z|$ that provides low reconstruction errors. This is the case for “vanilla” VAEs - the main object of this study, it is less so for AEs (Fig 8, CelebA) and fully connected VAEs (table 6, Fig 10, why not showing a figure like fig 7, 8?).
> > >
> > > So this begs the following question: a lot of work in the literature targets low reconstruction loss, or high generative quality. This is done by using some “sensible” $|z|$, and work on better modelling assumptions, such as hierarchical VAEs, normalising flows, and others.
> > >
> > > Why, in the authors’ opinion, working on $|z|$ is so important? Would it be possible to demonstrate that FONDUE could also improve the reconstruction performance of other variants of VAEs? This brings be to the second point.
> > >
> > > 2. It seems to me that this work is “restricted” to vanilla VAEs. Authors tried different architectures, and they also tried deterministic AEs, but in my view, it would be always possible for a reader to wonder if FONDUE could be applied to their preferred variant. How general is FONDUE?
> > >
> > > So, would it be fair to say that the ultimate quest is to find the ID of data, irrespectively of the model used to ingest the data and produce sensible output? In other words, as done by the authors in their experiments, for reconstruction purposes, the idea is to use the data ID as the “ground truth”, and check if FONDUE gets it right. So why wouldn’t be OK to just apply ID estimation on the data, and use this as a sensible choice for setting the latent dimension?
> > >
> > > To conclude, I am willing to rise my score, as I think the authors did a really good job in handling the rebuttal, and as mentioned in my original review (and by other reviewers) the proposed method has practical relevance. Nevertheless, I really am not sure this work meets the bar for an ICLR publication at this time. I hope the authors will continue improving their work, and find suitable venues for its publication.

---

> > > > ### Author Response · Authors · 2022-12-08
> > > > **Reply to reviewer EweH**
> > > >
> > > > > I would like [...] as read carefully the updated paper.
> > > >
> > > > Thank you very much for your time reading the updates.
> > > >
> > > > > Experiments with a simple downstream task [...] so whatever we chose as  seem to work.
> > > >
> > > > We agree that it would be interesting to see the results of FONDUE on more datasets, especially datasets where, like
> > > > Symsol, the results are sensitive to the dimensionality. We believe that the dimensionality choice should provide a good
> > > > tradeoff between reconstruction quality and downstream tasks performances, thus choosing the dimensionality at the "elbow"
> > > > of the reconstruction loss when no "elbow" is visible on the downstream tasks results.
> > > >
> > > > > This is the case for "vanilla" VAEs [...] figure like fig 7, 8?).
> > > >
> > > > Indeed, it seems that FONDUE will provide good reconstruction with deterministic AEs but not always the best performances
> > > > on downstream tasks, which will require more latent dimensions (on Symsol). We unfortunately forgot to update fig 10 in
> > > > the appendix on fully connected VAEs during our last update, hence the discrepancies with Fig. 7-8.
> > > >
> > > > > So this begs the following question: [...] general is FONDUE?
> > > >
> > > > Our usage of IDE is based on the polarised regime which has been shown to hold for VAEs optimising the ELBO and assuming a multivariate standard Gaussian prior.
> > > > Thus, as long as a variant of VAE learns in a polarised regime, FONDUE will work similarly to our example on "vanilla" VAEs.
> > > > It seems that the selected |z| can provide good reconstruction on other models (e.g., deterministic AEs) but further work would be needed to see if this
> > > > holds true for a larger variety of models, as mentioned in the conclusion.
> > > >
> > > > > So, would it be fair to [...] setting the latent dimension?
> > > >
> > > > Using the true data ID as the number of latent dimensions generally does not lead to good model performances.
> > > > For example, we know that the data IDs of Symsol and dSprites are 2 and 5, respectively,
> > > > but using these as the number of latent dimensions would lead to poor performances as can be seen in Fig 8.
> > > > Thus, simply setting $|z|$ as the data IDE would not provide good results, unless the overestimation corresponded
> > > > to a good dimensionality, which would be more by chance than by design.
> > > >
> > > > > To conclude [...] venues for its publication.
> > > >
> > > > We thank the reviewer for their thorough and insightful comments which greatly helped to improve the paper.

---

### Official Review · Reviewer_exVm · 2022-10-23

**Confidence:** 3
**Correctness:** 3
**Technical Novelty And Significance:** 3
**Empirical Novelty And Significance:** 2
**Recommendation:** 5

**Clarity, Quality, Novelty And Reproducibility:**

I find this paper to be interesting and potentially useful. The clarity on explaining some of the concepts, such as intrinsic dimensions, or polarized regime, could be a bit less hand-wavy. The experiments seem to be reproducible.

**Strength And Weaknesses:**

Strength:
1. A thorough experiments across a lot of settings for the intrinsic dimension estimations.
2. The experiments have averaged over random seeds which is more reproducible
3. The idea itself could be useful in determining latent dimension for VAE during training.

Weakness:
1. To make the paper more convincing, I think it should be discussed why one cannot just set the latent dimensions very big, such that it would be almost definitely contain the IDE. I understand the idea of "very big" is a bit vague, but I think to motivate the paper better, it should add discussion on if or why having significantly larger latent dimension than the IDE for VAE is less than ideal.
2. Another note, I think the word "intrinsic dimension" is often mixed up with the word "intrinsic dimension estimates", for example in conclusion. From my understanding, there should be only one intrinsic dimension for each dataset while the estimates could vary depending on the architecture.
3. I think the polarized regime is not very well explained. In dai-wipf's i believe some of the values in the encoder covariance go to 1 some to 0, thus polarized. But i think to simply say some collapse some don't, I'm not sure if it's really easy to understand for someone who hasn't read the original paper. i also think, in Dai-wipf's analysis, the intrinsic dimension is of the output diagonal covariance from the encoder, but here it seems to be considering the intrinsic dimension of a matrix, I suppose it's the layer that produces the covariance and mean. But then I am also confused as to what is the intrinsic dimension of a sampled representation.

**Summary Of The Paper:**

The paper explores intrinsic dimension estimation of the data using VAE. They found that this estimation can be made after a few steps of training and proposed a method FONDUE to provide a more principled method for selecting latent dimensions for the VAE.

**Summary Of The Review:**

I think this is an interesting paper but the clarity of writing and explanation could have some improvements. Also some discussion on the effect of having significantly higher latent dimension than intrinsic dimension should be discussed for a more well-rounded motivation.

---

> ### Author Response · Authors · 2022-11-11
> **Answer to reviewer exVm**
>
> Thank you for your interest in our work and for your detailed comments.
>
> Answers to Reviewer's Questions:
> --------------------------------
>
> > To make the paper more convincing [...] the IDE for VAE is less than ideal.
>
> This is an interesting point, we have added a paragraph in the introduction to discuss it.
>
> > Another note, I think the word "intrinsic dimension" [...] depending on the architecture.
>
> You are correct, we have updated the paper to differentiate between "intrinsic dimension estimates" (e.g., the results of MLE), and the "intrinsic dimension" or "true intrinsic dimension" (e.g. the real intrinsic dimension of a given dataset).
>
> >  I think the polarized regime is not [...] the original paper.
>
> We have now reformulated this paragraph and added more details in appendix F to clarify this.
>
> > i also think [...] of a sampled representation.
>
> We indeed use diagonal covariances as in Dai and Wipf. We use matrices for the IDE because we retrieve the layers activations for 10,000 data examples.
> For example, to estimate the ID of the mean representation, we use the matrix of activations of size $10,000 \times |z|$ given by the mean layer. Note that here, $|z|$ is the number of latent dimensions.
> The computation of the IDE of the sampled representation is similar and uses a matrix of size $10,000 \times |z|$ of sampled representations (one sample per data example).
> As this concept was also found unclear by reviewer psCP, we have now added some intuitive explanation and a toy example of ID estimation using Maximum Likelihood Estimation in the Appendix.
> The summary of our paper written by reviewer EweH also accurately describes how ID estimation is used in our paper.
>
> Other comments:
> ---------------
> > I think this is an interesting paper [...] more well-rounded motivation.
>
> Your interest in our work is very much appreciated, and we hope that the proposed updates clarify the paper and its motivation.

---

### Official Review · Reviewer_xYg3 · 2022-10-24

**Confidence:** 3
**Correctness:** 3
**Technical Novelty And Significance:** 2
**Empirical Novelty And Significance:** 3
**Recommendation:** 3

**Clarity, Quality, Novelty And Reproducibility:**

### Clarity

The paper is well-written and describes all the necessary steps.

### Quality

The paper is empirically motivated. All performed experiments are described in detail. It lacks experimental comparison to previous works.

### Novelty

As far as I know, the idea to use IDEs in VAEs is new. A related work section is missing, which makes assessing the novelty more difficult.

### Reproducibility

All the results seem to be reproducible. Code, dataset, and a lot of details are provided.

**Strength And Weaknesses:**

## Strengths
- Interesting and important topic
- Well written

## Weaknesses
- The ID calculation seems to be sensitive to the number of neighbors $k$ (as described in section 2.2). It is unclear how to set $k$ in practice for a dataset with no or limited access to ground truth data, and the proposed method does not sound too convincing either.
- No related Work in the main text. What about previous work that worked on automating the number of latent dimensions? It would also be interesting for experimental evaluation.
- I understand that the proposed method is mainly intended for VAEs as it requires the resampling operation. Are there extensions possible to go to a broader range of models? The extensions to autoencoder are a bit ad-hoc by saying we have an autoencoder as soon as we remove the resampling operation.
- Limited evaluation concerning potential downstream tasks. VAEs are often evaluated with respect to their ability to learn meaningful latent representations. It could be an interesting experiment to see the effect of a (semi)-automatic latent dimension specification on potential downstream tasks.


**Summary Of The Paper:**

The authors explore the intrinsic dimension estimation (IDE) of the data and latent representations learned by VAEs to measure the optimal size of the latent space.
They define the optimal dimension as having as many latent dimensions as possible but no passive measurements.
The authors evaluated their approach on three different datasets of increasing complexity.

**Summary Of The Review:**

Interesting paper, but the lack of related work in the main text and the missing comparison to baselines let weaknesses outweigh the interesting idea.

---

> ### Author Response · Authors · 2022-11-11
> **Answer to reviewer xYg3**
>
> Thank you for your interest in our work and for your detailed comments.
>
> Answers to Reviewer's Questions:
> --------------------------------
> > The ID calculation [...] (as described in section 2.2).
>
> Indeed, the ID estimation is sensitive to the number of neighbours and having a very precise IDE is still an ongoing topic of research.
> However, because our objective is to compare the IDEs of different representations, and not to accurately estimate their
> true ID, having stable IDEs is not less important than being as close as possible to the true ID, which is what
> guided our selection process for $k$, as explained in the answer to question 2.
> We have added additional results on FONDUE using MLE with k=3,5,10 in appendix I,
> showing that the way we selected $k$ generally leads to more consistent results which correspond to a good choice of
> dimensionality.
>
> > It is unclear how to set k [...] method does not sound too convincing either.
>
> We agree that the existing methods can seem unconvincing, but to the best of our knowledge, there is no principled method to select $k$ in the literature.
> As mentioned above, we tried to select the best possible $k$ based on a combination of multiple criteria to ensure stable results.
> Specifically we ensured that:
> - the amount of data used was sufficiently large (10,000 samples) and the selected $k$ provided a stable estimate over multiple runs, as recommended by [1]
> - the IDEs obtained with the selected $k$ were similar to the IDEs obtained with the closest value of $k$, as recommended by [2]
> - the IDEs obtained with the selected $k$ were generally similar to the IDEs provided by TwoNN.
>
> > No related Work in the main text. [...] experimental evaluation.
>
> We have now added [3,4] in the related work section as they propose some criteria to select the best number of latent dimensions
> when comparing multiple fully trained models. Specifically this is done using the elbow method (i.e., choosing the dimensionality corresponding to where the curve "bends") to select
> the number of dimensions based on the reconstruction loss or downstream task score.
> While these studies are not directly automating the selection of the number of latent dimensions, they were the closest contributions we could find despite a thorough search for related work.
> Additionally, we now discuss how FONDUE meets the criteria set by [3,4] in section 4.
> In an attempt to encourage more work on this question, in Sec. 5, we also discuss possible extensions to FONDUE that could be investigated.
>
>
> > I understand that the proposed method [...] we remove the resampling operation.
>
> While we have only checked how FONDUE could be used for autoencoders, it would be interesting to see if the provided dimensions could also be used to select the number of dimensions of
> other unsupervised models such as GANs. Despite leaving it for future work, we now discuss this question in the conclusion.
> The extension to autoencoders (AEs) is based on the results of Fig. 9, showing that the dimensionality provided by FONDUE
> leads to the same result as the elbow technique when monitoring the reconstruction loss (Note that we have also added a comparison with performance on downstream tasks).
> Our conclusions about AEs were thus not based on the discussion in Section 2.1 which was there only for the purpose of explaining the $\beta$ term of Eq.2.
> We initially did not present AEs in the background section due to space constraints, but have now added a short presentation of these models in appendix J to introduce them.
>
> > Limited evaluation concerning [...] downstream tasks.
>
> Thank you for your suggestion. We now discuss the F1 score of models with different latent dimensions on classification tasks in Sec. 4.
> This also allows us to compare the dimensions obtained by FONDUE with the results of [4] who selected the number of latent dimensions based on the accuracy on downstream tasks.
>
>
> Other comments:
> ---------------
> > Interesting paper, but [...] outweigh the interesting idea.
>
> Your interest in our work is very much appreciated. We have now added [3,4] in the related work and used their criteria to assess the quality of FONDUE's results in sec 4.
> We hope that these updates will address your concerns.
>
> References:
> -----------
> - [1] Elizaveta Levina and Peter J. Bickel. Maximum Likelihood Estimation of Intrinsic Dimension. In Advances in Neural Information Processing Systems, volume 16, 2004.
> - [2] Rasa Karbauskaite, Gintautas Dzemyda, and Edmundas Mazetis. Geodesic distances in the maximum likelihood estimator of intrinsic dimensionality. Nonlinear Analysis, 16(4):387–402, 2011.
> - [3] Carl Doersch. Tutorial on Variational Autoencoders. arXiv e-prints, 2016.
> - [4] Kien Mai Ngoc and Myunggwon Hwang. Finding the best k for the dimension of the latent space in autoencoders. In Computational Collective Intelligence, pp. 453–464. Springer International Publishing, 2020. ISBN 978-3-030-63007-2.

---

### Official Review · Reviewer_psCP · 2022-10-24

**Confidence:** 3
**Correctness:** 2
**Technical Novelty And Significance:** 2
**Empirical Novelty And Significance:** 2
**Recommendation:** 3

**Clarity, Quality, Novelty And Reproducibility:**

The paper is somewhat unclear. The method is closely related to methods that exist in the literature. The novelty seems to be a successive approximation of the intrinsic dimension based on calls to these methods (see comments). There are links to resources, but I am unsure if they guarantee reproducibility (since results related to some claims will be released upon acceptance).

**Strength And Weaknesses:**

### Strengths
+ Interesting problem will be useful to practitioners

### Weaknesses
- The organization of the paper is confusing (see comments)
- Some of the experiments and the notation in the proposed method are unclear (see comments)
- The contributions are not clearly laid out (see comments)
- As a result, it is difficult to assess the merits of the proposed work, and its contributions.


### Comments/Questions:
1. What is the motivation behind modeling a given data sample as distributed according to Poisson distribution in sec. 2.2? Why does this fit the dataset considered in the study?
2. What does the paper intend to communicate when it says "released more than 35000 IDE scores", what do these scores pertain to? does 35k refer to the number of models? Also, how are these counted?
3. Does FONDUE use the methods described in section 2.2? If yes, how does it use them, and why do the distributional assumptions apply (see #1)? If not, what does it rely on for IDE?
4. Related question to #3: What are IDE_z and IDE_mu. These appear to be scalars? How are these calculated n Algorithm 2? Do these rely on methods described in section 2.2? If so consider clearly defining these in the paper.
5. The organization is pretty confusing. While I like that the study is guided by experiments, and the authors use this to guide the proposed method. It is extremely unclear what the contributions are in terms of methodology.
6. Also, it seems FONDUE needs to train a VAE at every iteration of its execution? If would be useful to comment on the computational complexity of the algorithm. How does it scale with size of the dataset and the sample itself?

**Summary Of The Paper:**

The paper proposes an algorithm to estimate the intrinsic dimension of the latent representations of variational autoencoders. The algorithm is based on observational study based on other methods.

**Summary Of The Review:**

I do find the method to be of use in practice, and I am enthusiatic about it. It is just that the paper does not seem to be in good shape at the moment. I hope the authors can reorganize the paper and can clearly identify the technical contributions.

---

> ### Author Response · Authors · 2022-11-11
> **Answer to reviewer psCP**
>
> Thank you for your interest in our work and for your detailed comments.
>
> Answers to Reviewer's Questions:
> --------------------------------
>
> > 1.What is the motivation behind [...] in the study?
>
> We would like to emphasise that, in 2.2., it is the random variable $N(t, X_i)$ that is
> distributed according to the Poisson distribution, not the data sample.
> This distribution is used because $N(t, X_i)$ represents the number of points being within distance $t$ from $X_i$.
> For example, let us consider a 2-dimensional dataset with the set of points $X = {(0,0) (1,0) (0,1), (0,2)}$, and assume that
> they are the only data points in a radius 2 of $(0,0)$.
> We would then have $N(t=1, X_i=(0,0)) = 2$ and $N(t=2, X_i=(0,0))=3$ because
> we have 2 points at the Euclidean distance of 1 from $X_i=(0,0)$ and one at the Euclidean distance of 2 from $X_i=(0,0)$.
> We have now reformulated this description in the paper to avoid any confusion and added an intuitive explanation and a worked example of intrinsic dimension estimation using Maximum Likelihood Estimation (MLE) in Appendix H.
> We hope that this will help to make the paper more self-contained.
>
> > 2.What does the paper intend to [...] these counted?
>
> These scores represent the intrinsic dimension estimates (IDEs) of the 14 layers of the 5 runs of the VAEs considered and
> computed using 5 configurations (MLE with k=3, 5, 10, 20 and TwoNN) where the MLE was computed 3 times for each $k$, using different seeds.
> We computed these for:
> - 8 choices of latent dimensions for Symsol and dSprites, resulting in 16,800 IDEs
> - 14 choices of latent dimensions for Celeba, resulting in 14,700 IDEs
> - 1 choice of latent dimension with $\beta=20$ for dSprites, resulting in 1,050 IDEs
> - 1 choice of latent dimension at 1 early epoch for dSprites, Celeba and Symsol, resulting in 3,150 IDEs
> The total 35,700 IDEs were then used in Fig 2, 3, and 5.
> We have now added these details in Appendix B and reformulated IDE scores to IDEs for clarity.
>
> > 3.Does FONDUE [...] rely on for IDE?
>
> Yes, FONDUE uses the MLE technique presented in 2.2 to do intrinsic dimension estimation.
> After a few epochs (generally just one) of training a VAE, we will feed 10,000 data examples to the VAE and retrieve the corresponding mean and sampled representations.
> Using MLE with $k=20$, we then obtain the IDEs of the mean and sampled representations (i.e. $IDE_{\mu}$ and $IDE_z$) and compute the difference between them.
> See our answer to your question #1 regarding the distributional assumptions.
>
> > 4.Related question to #3: [...] consider clearly defining these in the paper.
>
> As defined in Theorem 1, $IDE_{\mu}$ and $IDE_z$ are the IDEs of the mean and sampled representations respectively.
> These are indeed scalars obtained using MLE with $k=20$ (see our answer to question #3 for more details).
> We now have added a paragraph which intuitively explains how FONDUE works to clarify this.
>
> > 5.The organization is [...] in terms of methodology.
>
> We have updated the list of contributions in the introduction of the paper and added some explanations in Sec. 4 to clarify this.
> Our main contribution is the FONDUE algorithm presented in 4.3, which, to the best of our knowledge, is the first
> algorithm to provide a good number of dimensions for the latent representations without requiring human supervision or full training of multiple models.
>
> Our other contribution is the study of the IDEs obtained at different layers of VAEs and at different steps of the training process (this has not been studied as far as we know).
> While this second contribution is minor, it was an important stepping stone to FONDUE and was thus presented in 4.2 as a logical introduction to FONDUE.
>
> > 6.Also, it seems FONDUE needs to [...] the sample itself?
>
> Let us first consider the computational complexity of our model, which should be $\mathcal{O}(e \times d \times p)$ where $e$ is the number of epochs, $d$ the number of data examples
> and $p$ the number of parameters of the model. Because we train the model for only a very low number of epochs, this is generally closer to $\mathcal{O}(d \times p)$.
> For simplicity, we will omit the time complexity of the IDE which is significantly smaller than the time complexity of training our VAE model.
> The computational complexity of FONDUE is thus $\mathcal{O}\left(n \times d \times p \times  log(n \times d \times p) \right)$ because we prune a huge part of the search space at each iteration $n$.
> Thus, what will really impact the time of execution of FONDUE is the number of data examples, the number of parameters of the model, and the number of iterations needed to reach the solution.
> Intuitively, FONDUE could be quite time-consuming for complex models using very large datasets, especially when the data IDE is large (i.e., the search space is larger, and we may need more iterations to reach convergence).
>
> (comments continue on next post)

---

> > ### Author Response · Authors · 2022-11-11
> > **Part 2 of comments**
> >
> >
> > Other comments:
> > ---------------
> >
> > > The method is closely related to methods [...] to these methods (see comments).
> >
> > We would like to emphasize that our goal is not to approximate the ID of the latent space, but to determine the number of latent dimensions z providing the best reconstruction and performances on downstream tasks.
> > Our paper shows that this non-trivial question can be addressed by FONDUE based on the difference between the IDEs of mean and sampled representations, which increases when the latent representations contain superfluous (unused) dimensions.
> > As mentioned in #5, to the best of our knowledge, FONDUE is the first algorithm to provide an estimate of the number of latent dimensions unsupervised and without fully training multiple models.
> > We hope that this will be clarified by the related work section introduced in the updated version of the paper.
> >
> > > I do find the method to be of use [...] the technical contributions.
> >
> > Your interest in our work is very much appreciated. We are sorry that you found the submitted version hard to follow,
> > and we hope that our answers and updates of the paper have made its content clearer.
> > While we have discussed here specific questions about ID estimation, we also recommend the very accurate summary of our paper written by reviewer EweH. They presented a good account of how FONDUE works, and what it does.

---

### Decision · Program_Chairs · 2023-01-20

**Decision:**

Reject

**Justification For Why Not Higher Score:**

No reviewer recommended to accept the paper.

**Justification For Why Not Lower Score:**

N/A

**Metareview: Summary, Strengths And Weaknesses:**

There was a consensus among reviewers that this paper should be rejected. While reviewers thought the paper had an interesting idea, the key concerns were that there as no comparison to alternative approaches and that there were no experimental results showcasing the approaches's usefulness for downstream tasks.